# Design of Plant-Based Food: Influences of Macronutrients and Amino Acid Composition on the Techno-Functional Properties of Legume Proteins

**DOI:** 10.3390/foods12203787

**Published:** 2023-10-15

**Authors:** Lena Johanna Langendörfer, Blerarta Avdylaj, Oliver Hensel, Mamadou Diakité

**Affiliations:** 1Faculty of Food Technology, University of Applied Science Fulda, Leipziger Str. 123, 36037 Fulda, Germany; blerarta.avdylaj@lt.hs-fulda.de (B.A.); mamadou.diakite@lt.hs-fulda.de (M.D.); 2Faculty of Organic Agricultural Science, University of Kassel, Nordbahnhofstraße 1a, 37213 Witzenhausen, Germany; agrartechnik@uni-kassel.de

**Keywords:** legumes, amino acid composition, correlation, techno-functional properties

## Abstract

Imitating animal-based products using vegetable proteins is a technological challenge that can be mastered based on their techno-functional properties. These properties of legume proteins can be influenced by multiple factors, among which the macronutrients and amino acid contents play an important role. Therefore, the question arises as to what extent the techno-functional properties are related to these factors. The water- and oil-holding capacities and the emulsion and foaming properties of commercially available legume protein powders were analyzed. Correlations between macronutrient, amino acid content, steric structure, and techno-functional properties were conducted. However, the protein concentration is the focus of techno-functional properties, as well as the type of protein and the interaction with the non-protein ingredients. The type of protein is not always quantified by the quantity of amino acids or by their spatial arrangement. In this study, the effects of the three-dimensional structure were observed by the used purification method, which overshadow the influencing factors of the macronutrients and amino acid content. In summary, both the macronutrient and amino acid contents of legume proteins provide a rough indication but not a comprehensive statement about their techno-functional properties and classification in an adequate product context.

## 1. Introduction

From 2018 to 2020, Europe’s plant-based food industry grew by 49% [1]. This remarkable increase is probably due to a shift in consumer awareness regarding the environment, health, and animal welfare, and the rising trend of veganism [2]. Added to this are the social, industrial, and political drives to achieve sustainable goals, such as the Sustainable Development Goals (SDGs). Moreover, to ensure the long-term nutrition of a continuously growing population and simultaneous sustainable food production to achieve climate targets, a shift in food consumption to plant-based products is of considerable relevance.

However, the switch from animal products to plant-based alternatives is still inhibited due to deficits in terms of product quality such as consistency, texture, appearance, etc. The techno-functional properties of raw materials can be used to make a statement about the general product properties and to enable classification in an adequate product context. They are influenced by various intrinsic and extrinsic factors [3,4,5,6].

According to Aryee et al. and Foegeding et al. [3,7], protein chemistry, in particular the quaternary structure and macronutrient content, seems to play an essential role with regard to techno-functional properties. However, the detection of the quaternary structure of proteins is a technical challenge because of its complexity. The quaternary structure is also dependent on the primary protein structure, about which the amino acid composition of a product can provide information [8]. In addition to the macronutrient composition, the amino acid composition is also often included in the product specifications. Therefore, it is of scientific interest to make statements about the techno-functional properties of the proteins based on their macronutrients and amino acid content.

All vegetable proteins have the potential to serve as the basis for a plant-based alternative. Due to the high protein content and technological and functional properties such as viscosity, water and oil absorption capacity, foaming, and emulsification, legumes and their processed products are frequently used in foods [9,10]. Most plant-based products are based on soy protein or soy starch [11]. However, soy as an ingredient often leads to unsatisfactory texture and it shows allergic potential [12,13,14]. Furthermore, soybean production leads to deforestation and has a high carbon footprint [15]. Due to this, other legumes are preferred in the plant-based industry [11].

Several studies deal with the techno-functional properties and/or amino acid composition of legume proteins. Kinsella [6] and Shresta et al. [16] summarized the state of the literature regarding the major globular components and the techno-functional properties of soybean and lupine. The latter authors also focused on the extraction methods used and their influence on the techno-functional properties. This was also the focus of Lam et al. [5] for pea protein isolate and Sharan et al. [17] for fava bean. Furthermore, Klupsaite et al. [18] and Keskin et al. [19] compared legumes in terms of their techno-functional properties.

Iqbal et al. [20] compared the mineral constitution and amino acid profile of four important legumes, and Sosulski et al. [21] investigated the amino acid composition of soy meal and protein isolates. Gorissen et al. [22] analyzed the EAA (essential amino acid) and general amino acid contents of a large selection of plant-based proteins. All of these research studies focused mainly on the nutritional value of the amino acids, but not on their impact on the techno-functional properties. Therefore, fewer studies deal intensively with the relationship between amino acids, macronutrient composition, and techno-functional properties. 

Due to this, the following questions arise: How do the amino acid and macronutrient compositions affect the techno-functional properties of legume proteins? How relevant are the techno-functional properties to the spatial structure of the proteins? Can the amino acid and macronutrient compositions be used to make predictions about the techno-functional properties and suitable product applications? These and further questions are investigated in more detail in this study.

## 2. Materials and Methods

### 2.1. Materials

Both the protein isolate and concentrate were investigated in this research. Pea protein isolate (PPI; Pisane M9, Cosucra) and lupin protein isolate (LPI, lupin protein isolate spray dried Pro Lupin), chickpea protein concentrate (CPC; debitterized, VIRIDI Foods GmbH), and sunflower oil (Chemiekontor.de GmbH) were purchased. Herba ingredients kindly provided the PPC (pea protein concentrate) and FPC (fava bean protein concentrate).

Based on the specification of the manufacturers, Table 1 shows the macronutrient information (carbohydrate, fiber, fat, and protein).

The amino acid compositions of the PPI, LPI, and CPC are based on the manufacturers’ information, while the amino acid compositions of the PPC and FPC are according to an analysis by SGS (Institut Fresenius) (Table 2).

### 2.2. Methods

In this study, laboratory tests were conducted to determine the techno-functional properties of the legumes.

#### 2.2.1. Water-Holding Capacity (WHC) and Oil-Holding Capacity (OHC)

These were determined using the method of Beuchat [23], with slight modification. In a centrifuge tube, 50 mL of a 3.5% wt protein suspension was prepared with water or oil using an Ultra Turrax: IKA T18 basic (IKA, Staufen, Germany, dispersing tool S1N-19G). Afterwards, the sample was centrifuged at 4000 rpm for 5 min at 20 °C. The supernatants obtained were decanted, and the centrifuge tubes containing sediment were weighed. Then, the remaining mass of water/oil per gram of sample was calculated.

#### 2.2.2. Preparation of the Emulsion

A homogenous protein dispersion (3.5% wt) was prepared in a 100 mL beaker with an Ultra Turrax: IKA T18 basic (dispersing tool S1N-19G). Then, 10 g of sunflower oil was added, and the mixture was emulsified for one minute.

#### 2.2.3. Oil Volume Fraction of the Emulsion (Φ)

The oil content of emulsion was determined according to Pearce and Kinsella [24].

#### 2.2.4. Emulsion Activity Index (EAI)

The EAI was analyzed by the turbidimetric technique of Pearce and Kinsella [24], with slight modifications. From the bottom of the beaker, 0.05 mL of emulsion was removed and diluted (1/101) with a 0.01% SDS solution. Then, the absorbance was measured at 500 nm in a UV–Vis spectrophotometer. The EAI is calculated as follows:(1)$$\mathrm{EAI}\;\left[\frac{m^{2}}{g}\right]=\frac{2\cdot 2.303\cdot A\cdot \mathrm{VF}}{l\cdot c\cdot \;\Phi }$$
where A = absorption; VF = dilution factor; l = layer thickness of the cuvette (m); c = protein concentration (g/m^3^); Φ = oil volume fraction of the emulsion (-).

#### 2.2.5. Emulsion Stability (ES)

The ES shows the percent of EAI existing after 10 min at room temperature.

#### 2.2.6. Foam Capacity (FC)

The FC or whippability was characterized with the method of Watanabe et al. [25], with some minor adjustments to the laboratory equipment. From a homogeneously prepared protein suspension (3.5% wt), 20 mL was transferred to a 100 mL beaker. Then, the mixture was foamed for 20 s using a milk frother. The content of the beaker was directly transferred into a measuring cylinder, where the foam volume was visually determined.
(2)$$\mathrm{FC}\;\left[\%\right]=\frac{V_{\mathrm{foam}}}{V_{a}}\times \;100$$
where V_foam_ = foam volume; V_a_ = initial volume

#### 2.2.7. Foam Stability (FS)

After 60 min at room temperature, the foam volume was determined again, and the percentage of remaining foam was expressed as FS.

#### 2.2.8. Amino Acid Classifications

In Table 3, the amino acids are classified according to their steric properties.

In Table 3, the mentioned groups were calculated with the summation of the amino acids in g/100 g (Table 2).

#### 2.2.9. Statistical Analysis

The WHC/OHC measurements were performed eight times, and the remaining techno-functional properties were determined three times. One-factor ANOVA and the Tukey–Kramer post hoc test (*p* < 0.05) were applied using MATLAB R2022a to analyze significant differences in the techno-functional properties.

In addition to the techno-functional properties of the different legumes, this study is particularly concerned with the influence of the intrinsic factors of macronutrient and amino acid compositions on them. In this context, correlations were examined in order to make preliminary statements on the techno-functional properties based on the information on the macronutrient and amino acid compositions. For this, the Pearson correlation coefficient and the *p*-values were calculated using MATLAB R2022; a result was considered significant if *p* < 0.05.

## 3. Results

### 3.1. Techno-Functional Properties

#### 3.1.1. Water- and Oil-Holding Capacities (WHC/OHC)

The isolates showed higher WHC values than the concentrates, the PPI showing the highest significant value (5.89 g/g (±0.22) followed by LPI with (2.45 g/g (±0.17) (see Figure 1). Among the concentrates, CPC (1.54 g/g (±0.26) and PPC (1.31 g/g (±0.09) showed significantly higher WHC than FPC (1.03 g/g (±0.07). Compared to the WHC, the OHC of the individual raw materials showed results that were more similar. The PPC (2.04 g/g (±0.09), LPI (2.56 g/g (±0.64), and FPC (1.72 g/g (±0.30) showed a higher OHC than WHC, although this difference was not significant for the LPI.

#### 3.1.2. Emulsion Activity Index (EAI) and Emulsion Stability (ES)

The significantly highest EAI (7.58 m^2^/g (±0.17) was from the PPC, which was followed by the PPI (6.11 m^2^/g (±0.37) and LPI (5.55 m^2^/g (±0.22) (see Figure 2). The FPC and CPC showed lower values (4.75 m^2^/g (±0.34) and (5.01 m^2^/g (±0.13). In terms of emulsion stability, the PPI (92.84% (±1.43) and FPC (95.52% (±1.68) showed the most significantly stable emulsions. The PPC (86.87% (±2.14) and the LPI (86.77% (±0.97) followed. The most significant unstable emulsion was the CPC (73.27%) (±1.27).

#### 3.1.3. Foam Capacity (FC) and Foam Stability (FS)

In terms of their foam-forming properties, the LPI showed by far the highest significant FC (225.00% (±5.00) (see Figure 3). In contrast, all of the other proteins showed less than half of that. For example, the PPI had an FC of 116.67% (±18.76), followed by the FPC with 50.00% (±5.00). Furthermore, the PPC had 22.50% (±6.61) and CPC had 17.50% (±2.50), with no significant differences. Regarding foam stability, no significant differences were evident (90.74–65.70%) except for the PPC, which had the lowest value (30.36% (±6.44).

### 3.2. Correlations of Macronutrients, Amino acid Composition, Steric Structure, and Techno-Functional Properties

#### 3.2.1. Correlations of Macronutrients and Techno-Functional Properties

Considering the relationships between the macronutrients and techno-functional properties, some correlations stand out (Table 4). There was one significant positive correlation: PC and FC with 0.95. The FS and EAI were significantly negatively correlated (−0.91). Negative correlations were also shown between ES and fiber (−0.95) and between FC and fat (−0.91). Insignificant correlations higher than or similar to 0.80 were found between the C and FC (−0.81), and between the FC and OHC (0.80).

#### 3.2.2. Correlations between Techno-Functional Properties and Each Amino Acid

Some amino acids show high linear correlations with techno-functional properties (Table 5), such as the significant correlation between OHC and Ala (−0.94), Pro (−0.91), Tyr (0.97), and Val (−0.88). The FC and Ala (−0.92), Asp (−0.97), Glu (0.88), Lys (−0.89), and Thr (−0.92) were significantly related. Concerning the ES, Ile influenced it significantly in a negative manner (−0.91). Furthermore, high but insignificant relations were shown between Arg and FC (0.82), Cys, Gly, Leu, and WHC (−0.80), (−0.87), and (0.83), respectively, and between Val and FC (−0.80).

#### 3.2.3. Correlations between Techno-Functional Properties and Amino Acid Groups According to Steric Structure

Concerning the correlations between techno-functional properties and amino acid composition classified according to the steric structure (Table 6), only one significant correlation was found: OHC and −ve (0.88). The positive amino acids also correlated highly and negatively with the OHC (−0.80). Moreover, the FC and H, ES and P, and ES and S also showed negative relations (−0.82, −0.82, and −0.84, respectively).

## 4. Discussion

### 4.1. Techno-Functional Properties

The techno-functional properties are influenced by various intrinsic and extrinsic factors. In addition to the intrinsic factors such as the variety, origin, amino acid structure, and purification method, the conditions during measurement (pH, temperature, protein concentration, etc.) as well as the performance of each method have a major influence on the techno-functional properties. For example, Barac et al. [26] showed an EAI variation of 40–260 m^2^/g at pH values between 3 and 8 within a genotype. Therefore, the results of the techno-functional properties should always be related to the measurement conditions and the measurement method. Generalizations concerning the plant variety are very limited, because every result is specific to the protein raw material and the specific production method [4,27,28,29,30,31,32,33].

#### 4.1.1. Water- and Oil-Holding Capacities (WHC/OHC)

According to Wang and Kinsella, as well as Zayas [6,34], the WHC tends to increase by increasing the protein concentration, which is confirmed by the results because the isolates showed a higher WHC than the concentrates.

The quantitative number of hydrophobic vs. hydrophilic amino acids alone cannot explain the high WHC of PPI. However, the spatial structure could be decisive here, which is strongly influenced by factors such as the purification method (see Section 4.2.2 below). Probably, the hydrophilic amino acids in PPI are mostly exposed in comparison to LPI. The hydrophilic amino acids can better interact with water which leads to higher WHC [35]. This can explain the highest significant value of the PPI. Compared to other legume protein isolates, Alu’datt et al. [36] also determined lower WHC values for lupin.

In addition to the protein content and the spatial structure, the non-protein components also play a relevant role with regard to the techno-functional properties. CPC has by far the highest fiber content, followed by PPC and FPC, which have the lowest. Fibers from legumes also have a good ability to bind water, whereby soluble fractions, such as pectic substances, increase the WHC [19,37]. Due to the generally higher fiber content, CPC and PPC probably also contain more soluble fractions, which increase the WHC in contrast to FPC. The WHC of PPC and CPC do not show significant differences, although CPC has by far a higher fiber content. PPC contains a higher protein content and the spatial protein structure could also favor the WHC, which would make the high fiber content less significant.

The higher OHC values for PPC, LPI, and FPC probably occur because these proteins show good hydrophobicity, and the aliphatic chains of the lipids can interact better with the nonpolar side chains of the amino acids than water with the polar sides [38].

#### 4.1.2. Emulsion Activity Index (EAI) and Emulsion Stability (ES)

The emulsion properties (EAI, ES) of proteins are particularly dependent on two factors: firstly, the rate at which the protein diffuses to the interface, and secondly, the deformability of its conformation under the influence of the interfacial tension (surface denaturation) [39].

Legumes consist mainly of globular proteins, which generally have poorer emulsification properties due to their compact and inflexible structure [40]. Globular proteins can partially unfold due to heating above their denaturation temperature and shearing [41,42].

At lower protein concentrations, the protein can better unfold during shearing [41]. Furthermore, high protein concentrations can lead to high activation energy barriers, which impede the migration of proteins [43]. This explains the high EAI of PPC in contrast to PPI and LPI. However, despite their lower protein content, FPC and CPC do not have a higher EAI than PPI and LPI.

In addition to the quantitative protein content, the type of protein also probably plays a major role in relation to EAI. The proteins in legumes consist mainly of globulins and albumins. In terms of globulins, the major proteins are legumin (11S) and vicilin (7S) [44]. According to Dagorn-Scaviner et al., the vicilin–legumin ratio influences the emulsifying properties, whereby vicilin shows in legumes better emulsifying properties than legumin. Probably, vicilin has lower molecular weight, a more flexible tertiary structure, and the proneness to carrying glycosylated groups, leading to better emulsification than legumin [45].

As claimed by Gravel et al. [46], pea can contains the highest vicilin amount, followed by lupin, fava bean, and chickpea. This could lead mainly to a higher EAI of the pea powders and LPI in comparison to the lower values for FPC and CPC.

Emulsions are thermodynamically unstable because of the increased interfacial free energy of the system. Since the system strives to minimize the free energy, an emulsion is subject to creaming, flocculation, and coalescence after a certain time [5].

According to Barac et al., coactive effects, such as high solubility, protein composition, and structure, influence ES [26]. The interaction of all coactive effects lead to the high ES values of FPC and PPI followed by PPC and LPI. The statistically lowest ES of CPC in comparison to the other legumes may be due to the interaction of the low protein and the high fat content, because an increased protein concentration can lead to the formation of smaller oil droplets and fat can disrupt emulsion stability [47,48,49].

#### 4.1.3. Foam Capacity (FC) and Foam Stability (FS)

In general, albumins show better foaming capacity and stability than globulins [50]. Lupin protein has a high albumin content (25%) compared to other proteins, which explains its good FC [10]. Moreover, the literature also shows that lupin protein (5–2083%) can achieve a higher FC than, for example, pea protein (9–263%) [4,25,27,29,30,31,51].

The isolates have a significantly higher FC than the concentrates. A higher protein concentration does not always lead to a higher FC [49]. Proteins have an individual concentration threshold until the FC increases; then, it decreases [52]. The significantly lower FC of CPC could be due to its comparatively higher fat content, since lipids displace proteins from the gas surface, due to their hydrophobicity, without being able to form stable films themselves [34,39].

Foams collapse as larger bubbles grow at the expense of smaller bubbles (disproportionation). This means that the structure and density of the foam also influence the foam’s stability. The FC indicates the surcharge volume, but it does not apprise the bubble size distribution. Proteins with a high FC do not necessarily show a high FS, as the latter depends mainly on intermolecular interactions, cohesiveness, protein film strength, and gas permeability [31]. Therefore, a protein may have a low FC but a better FS. CPC shows a low foam volume, but it has a finer pore foam with strong protein films.

### 4.2. Correlations of Macronutrients, Amino Acid Composition, Steric Structure, and Techno-Functional Properties

#### 4.2.1. Correlations between Macronutrients and Techno-Functional Properties

As mentioned in the previous section, the protein concentration can positively influence the FC until an individual threshold is reached, which in this case was probably not reached, as there is a strong positive correlation between the PC and FC [40,52].

Although the basic principles of forming and stabilizing a foam and an emulsion are similar, the interfaces have different energetics, which means that the molecular requirements are also different [35]. Thus, proteins with good emulsifying properties are not equally good foam formers and also negative relations between these properties are possible.

According to Taherian et al., Belitz et al., and Bandyopadhyay et al., hydrocolloids or polysaccharides tend to improve emulsion stability. They increase the viscosity of the outer aqueous phase and due to protein–polysaccharid complexes, increase the thickness of the liquid between two closely spaced droplets [32,39,53].

The results of this study could not confirm the positive influence. In this work, a negative correlation was measured. Factors such as biopolymer (size, concentration, and type), solvent conditions (temperature, pH, and salts) and methods of emulsion preparation influence the function of the protein polysaccharide complex [54,55,56,57]. There is probably an unfavorable combination here, which explains the negative correlation.

As already mentioned, fats interfere with foam formation, which is again confirmed by this strong negative correlation between FC and fat.

According to Damodaran [35], the addition of low molecular weight carbohydrates improves the whipping of whey proteins, but in the case of higher molecular weight carbohydrates such as starch, there is a tendency to decrease it. Sugars can also influence the FC negatively because they enhance the stability of proteins, which leads to possible inhibition of unfolding upon adsorption at the interface [35]. Moreover, carbohydrates show no affinity for the air–water interface [35]. This could be the reason for the negative correlation between carbohydrates and foam capacity.

Both the FC and OHC are improved by a protein with a high surface hydrophobicity, which could explain the positive correlation between the two techno-functionalities [5,35,39].

#### 4.2.2. Correlations of Techno-Functional Properties and Each Amino Acid

The amino acids Ala and Pro can lead to an exposure of hydrophobic groups in the core, because of their short side chains and their ring structures [8,58,59]. However, these amino acids show a significant negative correlation with the OHC, although oil interacts better with the hydrophobic parts of a protein [3,5,38].

This correlation is probably due to the comparatively high OHC and low Ala and Pro contents of LPI.

This and the other following contradictory results are probably due to the quaternary structure of the protein and are not only influenced by the primary structure or amino acid profile. The spatial structure is also particularly dependent on the type of purification method and its prevailing conditions (e.g., pH values, salt concentration, or even temperature), which in turn influence the techno-functional properties of the proteins.

This is confirmed by several literature sources, e.g., Stone et al. [27], which demonstrated significant effects of different purification methods (alkali extraction/isoelectric precipitation (AE-IP), salt extraction–dialysis (SE), and micellar precipitation (MP) on the techno-functional properties (WHC, OHC, and foaming properties) of pea protein.

Rodriguez et al. [31] also measured differences in isoelectric precipitation and micellization between soy and lupin with respect to their WHC, OHC, and foaming properties.

Hu et al. [33] and Joshi et al. [60] analyzed the influence of the drying method on the techno-functional properties. The former authors found better emulsifying and foaming properties for spray-dried SPIs compared to freeze-dried and vacuum-dried SPIs. The latter authors found a lower water absorption capacity in spray-dried lupin powders compared to freeze-dried and vacuum-dried powders.

In addition to the type of production, the process itself also influences the techno-functional properties. Berghout et al. [28] investigated the impact of different process parameters of the used purification method on the techno-functional properties of lupine protein.

Due to the polar nature of Tyr, it should have a negative influence on the OHC. Nevertheless, the results showed a positive correlation, which is also due to the comparatively high OHC and high tyrosine content of lupine.

Furthermore, Val has a significant negative effect on the OHC and FC because Val may enhance the hydrophobic effect due to the high hydrophobicity of its side chains. The hydrophobic effect is the main force that leads to a hydrophobic interior, impeding the access of the aliphatic side chains of the lipids to the hydrophobic amino acid side chains [61].

As with the OHC, the negative correlation between FC and Ala is particularly affected by the result of the LPI. Furthermore, Thr, Asp, Arg, and Lys show a negative correlation with FC; these amino acids are charged and polar. However, a small net charge and high surface hydrophobicity are conducive to a high FC, which these amino acids can counteract [5,35,39].

The positive connection between FC and Glu is mainly because albumins contain glutamic acid and improve the foaming properties [62].

Gly can break the compact structure of the hydrophobic core due to the missing side group, and thus expose the hydrophobic internal of the globular protein, which can negatively influence the interaction with water or the WHC [63].

Furthermore, Val and Ile also have a high hydrophobicity, which can also have a negative influence on the ES [61].

The amino acid Cys is a hydrophilic amino acid whereas Leu is hydrophobic. Cys should bind water well and Leu should bind water poorly [8,64]. However, this is not reflected in the results with correlations between WHC and Cys (−0.80) and Leu (0.83). This can be explained by the fact that PPI shows, in comparison, a very high WHC, although it shows the lowest Cys and the highest Leu content. As already mentioned in Section 4.1.1, here, the arrangement in the tree-dimensional structure plays a superior role in comparison to the quantity of amino acids.

#### 4.2.3. Correlations between Techno-Functional Properties and Amino Acid Groups According to Steric Structure

The OHC is strongly and negatively influenced by positively and negatively charged amino acids. Oil interacts with the non-polar side chains of amino acids, and the charged side chains can negatively influence oil binding [3,5,34].

The negative impact of the hydrophobic side chains on the FC can be justified by the same argument as for the negative influence of valine. Not only is the hydrophobicity/hydrophilicity amino acid ratio of proteins the primary determinant of surface activity, the distribution pattern of hydrophilic and hydrophobic groups on the protein surface is decisive [35]. Although the amino acid composition influences the three-dimensional structure of the protein, other factors also alter it.

The strength of a protein film depends on cohesive intermolecular interaction. This includes attractive electrostatic interactions, hydrogen bonding, and hydrophobic interactions. Probably, the negatively charged amino acids negatively influence electrostatic interactions, resulting in poorer emulsion stability [35].

Amino acids with sulfide groups support the viscoelastic properties of the protein film due to the polymerization of adsorbed proteins by disulfide–sulfhydryl interchange reactions. This leads to a more stable emulsion, whereas the study’s results show the opposite [35]. The contradictory result is probably due to the high proportion of amino acids with sulfide groups, and the low ES of the CPC. Instead of sulfide groups, other factors may be involved in the low ES.

## 5. Conclusions

The aim of this research was to analyze the relationship between amino acid/macronutrient composition and techno-functional properties of different legumes. According to the results, the amino acid profile and the macronutrient content affect the techno-functional properties of legumes, which is illustrated by some significant correlations.

Furthermore, the role of the quaternary structure should be examined more closely in this context. Not all correlations can be attributed to the two influencing factors (amino acid/macronutrient composition). This could be mainly due to the fact that the quaternary structure affects the techno-functional properties of proteins. The spatial structure is not only influenced by the primary structure; the type and process of the purification method used also show a great impact.

Nevertheless, the macronutrient and amino acid compositions can help to make statements about the techno-functional properties of legume proteins in advance, which in turn facilitates their classification in an adequate plant-based product context.

Regarding the emulsion properties, in addition to the measured parameters, measurements of surface hydrophobicity, percentage of adsorbed protein, and interfacial protein concentration would help to describe more specifically the emulsion properties of the protein powders. Furthermore, to be able to develop the correlations even more precisely, other techno-functional or physicochemical properties (solubility, gelling properties, denaturation properties, zeta potential, surface hydrophobicity, interfacial tension, etc.) should also be taken into account. These properties are relevant for product design, especially about the texture formation, heat processing, and interface properties, and would promote the classification of legume proteins in a product-based context. Other plant-based raw materials can be determined to describe differences between plant families.

## Figures and Tables

**Figure 1 foods-12-03787-f001:**
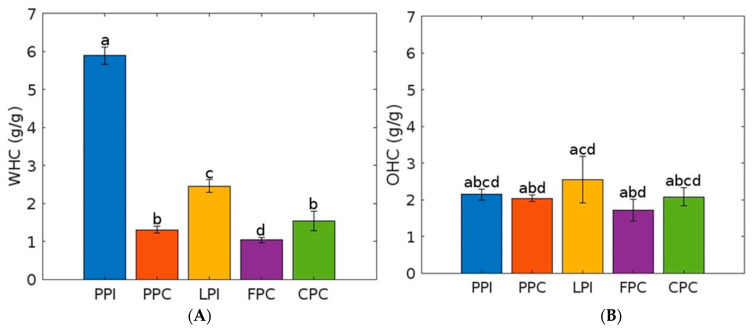
Water-and oil holding capacity of protein powders. PPI = pea protein isolate, LPI = lupin protein isolate, PPC = pea protein concentrate, FPC = fava bean protein concentrate, CPC = chickpea protein concentrate. (**A**) WHC = water holding capacity, (**B**) = oil holding capacity. Error bars showing standard deviation. Different letters indicate significant differences (*p* < 0.05) in the measured parameters.

**Figure 2 foods-12-03787-f002:**
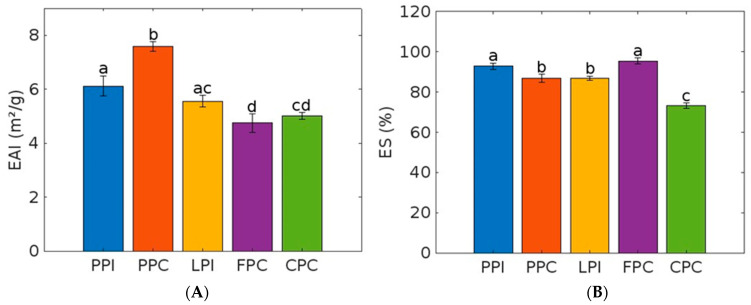
Emulsion properties of protein powders. PPI = pea protein isolate, LPI = lupin protein isolate, PPC = pea protein concentrate, FPC = fava bean protein concentrate, CPC = chickpea protein concentrate. (**A**) EAI = emulsion activity index, (**B**) ES = emulsion stability. Error bars showing standard deviation. Different letters indicate significant differences (*p* < 0.05) in the measured parameters.

**Figure 3 foods-12-03787-f003:**
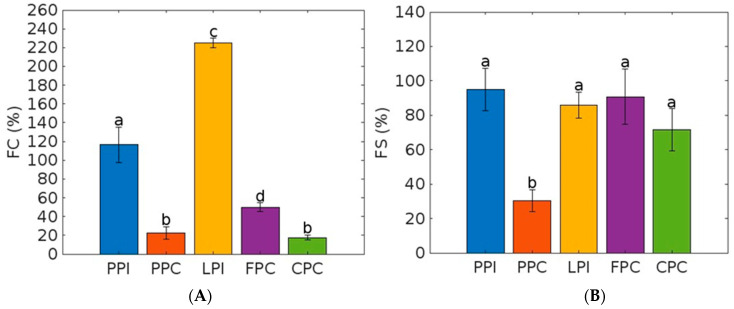
Foaming properties of protein powders. PPI = pea protein isolate, LPI = lupin protein isolate, PPC = pea protein concentrate, FPC = fava bean protein concentrate, CPC = chickpea protein concentrate.168 (**A**) FC = foam capacity, (**B**) FS = foam stability. Error bars showing standard deviation. Different letters indicate significant differences (*p* < 0.05) in the measured parameters.

**Table 1 foods-12-03787-t001:** Macronutrient amount of the protein powders in g/100 g product.

Macronutrient	PPI	PPC	LPI	FPC	CPC
Carbohydrate (C)	0.80	18.50	0.50	5.78	18.00
Fiber (FI)	2.40	5.78	4.40	3.47	15.80
Fat (FA)	4.00	6.94	3.00	6.36	9.00
Protein (PC)	81.70	47.58	91.00	57.85	42.00

PPI = pea protein isolate, LPI = lupin protein isolate, PPC = pea protein concentrate, FPC = fava bean protein concentrate, CPC = chickpea protein concentrate.

**Table 2 foods-12-03787-t002:** Amino acid compositions of the protein powders in g/100 g protein, as is.

Amino Acids	PPI	PPC	LPI	FPC	CPC
Ala	4.30	4.42	3.37	4.70	4.51
Arg	8.70	8.60	11.14	9.76	8.25
Asp	11.50	12.18	11.02	11.77	11.98
Cys	1.00	1.36	1.35	1.25	1.38
Glu	16.80	17.04	23.73	17.83	16.31
Gly	4.10	4.36	4.16	4.32	4.20
His	2.50	2.52	2.59	2.74	2.89
Ile	4.50	4.40	4.39	4.25	5.93
Leu	8.40	7.69	7.54	7.83	7.74
Lys	7.20	8.05	4.39	6.98	7.03
Met	1.10	1.01	0.56	0.77	1.28
Phe	5.50	5.41	4.16	4.58	5.02
Pro	4.50	4.42	4.27	4.58	4.41
Ser	5.30	5.12	4.95	5.25	5.33
Thr	3.90	3.94	3.37	3.84	4.04
Trp	1.00	0.94	0.90	0.89	1.09
Tyr	3.80	3.77	4.16	3.58	3.71
Val	5.00	4.76	3.94	5.09	4.91

PPI = pea protein isolate, LPI = lupin protein isolate, PPC = pea protein concentrate, FPC = fava bean protein concentrate, CPC = chickpea protein concentrate, alanine (Ala), arginine (Arg), aspartic acid (Asp), cysteine (Cys), glutamic acid (Glu), glycine (Gly), histidine (His), isoleucine (Ile), leucine (Leu), lysine (Lys), methionine (Met), phenylalanine (Phe), proline (Pro), serine (Ser), threonine (Thr), tryptophan (Trp), tyrosine (Tyr), valine (Val).

**Table 3 foods-12-03787-t003:** Amino acid classification according to steric structure [8].

H	OH	P	+ve	−ve	A	S
Ala	Ser	Ser	Lys	Asp	Trp	Met
Gly	Thr	Thr	Arg	Glu	Phe	Cys
Trp	Tyr	Tyr	His			
Phe		Cys				
Val						
Leu						
Ile						
Met						
Pro						

H = hydrophobic, OH = polar amino acids with OH groups, P = polar amino acids, +ve = positively charged amino acids, −ve = negatively charged amino acids, A = aromatic amino acids, S = amino acids with sulfur groups, alanine (Ala), arginine (Arg), aspartic acid (Asp), cysteine (Cys), glutamic acid (Glu), glycine (Gly), histidine (His), isoleucine (Ile), leucine (Leu), lysine (Lys), methionine (Met), phenylalanine (Phe), proline (Pro), serine (Ser), threonine (Thr), tryptophan (Trp), tyrosine (Tyr), valine (Val).

**Table 4 foods-12-03787-t004:** Correlations between functional properties and macronutrients.

	WHC	OHC	EAI	ES	FC	FS	C	FI	FA	PC
WHC	1.00	0.33	0.14	0.30	0.42	0.00	−0.60	−0.40	−0.59	0.63
OHC		1.00	0.16	−0.28	**0.80**	0.05	−0.35	−0.01	−0.59	0.65
EAI			1.00	0.09	−0.12	**−0.92**	0.32	−0.25	−0.09	−0.07
ES				1.00	0.26	0.16	−0.64	**−0.95**	−0.60	0.47
FC					1.00	0.45	**−0.81**	−0.48	**−0.91**	**0.95**
FS						1.00	−0.62	−0.07	−0.28	0.41

WHC/OHC = water-holding capacity/oil-holding capacity, EAI = emulsion activity index, ES = emulsion stability, FC = foam capacity, FS = foam stability, C = carbohydrate content as is, FI = fiber content as is, FA = fat content as is, PC = protein content as is. Correlations higher than 0.80 are in bold numbers.

**Table 5 foods-12-03787-t005:** Influence of amino acid content on techno-functional properties based on correlation analysis.

	Ala	Arg	Asp	Cys	Glu	Gly	His	Ile	Leu	Lys	Met	Phe	Pro	Ser	Thr	Trp	Tyr	Val
WHC	−0.22	−0.10	−0.44	**−0.87**	−0.05	**−0.80**	−0.53	−0.15	**0.83**	−0.05	0.18	0.41	0.06	0.21	−0.05	0.21	0.25	0.06
OHC	**−0.94**	0.49	−0.70	0.14	0.73	−0.59	−0.32	0.02	−0.25	−0.74	−0.33	−0.33	**−0.91**	−0.66	−0.71	−0.02	**0.97**	**−0.88**
EAI	−0.06	−0.29	0.32	0.00	−0.14	0.29	−0.74	−0.32	0.06	0.41	0.13	0.60	−0.18	−0.29	0.13	−0.09	0.17	−0.09
ES	0.04	0.36	−0.28	−0.61	0.13	0.10	−0.60	**−0.91**	0.40	0.02	−0.53	−0.04	0.52	−0.13	−0.25	−0.74	−0.10	0.14
FC	**−0.92**	**0.82**	**−0.97**	−0.18	0.88	−0.61	−0.42	−0.41	−0.07	**−0.89**	−0.71	−0.58	−0.58	−0.68	**−0.92**	−0.44	**0.88**	**−0.80**
FS	−0.22	0.60	−0.63	−0.10	0.44	−0.38	0.45	0.00	−0.06	−0.65	−0.46	−0.76	0.06	0.00	−0.46	−0.21	0.10	−0.16

WHC/OHC = water-holding capacity/oil-holding capacity, EAI = emulsion activity index, ES = emulsion stability, FC = foam capacity, FS = foam stability, alanine (Ala), arginine (Arg), aspartic acid (Asp), cysteine (Cys), glutamic acid (Glu), glycine (Gly), histidine (His), isoleucine (Ile), leucine (Leu), lysine (Lys), methionine (Met), phenylalanine (Phe), proline (Pro), serine (Ser), threonine (Thr), tryptophan (Trp), tyrosine (Tyr), valine (Val). Correlations higher than 0.80 are in bold numbers.

**Table 6 foods-12-03787-t006:** Correlations between functional properties and steric structure of amino acids.

	H	OH	P	+ve	−ve	A	S
WHC	0.13	0.31	−0.26	−0.44	0.33	0.55	0.01
OHC	−0.63	−0.32	−0.22	**−0.80**	**0.88**	0.05	0.23
EAI	0.04	0.11	0.10	0.17	0.46	0.67	0.36
ES	−0.25	−0.44	**−0.82**	0.57	−0.49	−0.18	**−0.84**
FC	**−0.82**	−0.64	−0.73	−0.52	0.45	−0.31	−0.38
FS	−0.36	−0.41	−0.46	−0.19	−0.37	−0.77	−0.60

WHC/OHC = water-holding capacity/oil-holding capacity, EAI = emulsion activity index, ES = emulsion stability, FC = foam capacity, FS = foam stability, H = hydrophobic, OH = polar amino acids with OH groups, P = polar amino acids, +ve = positively charged amino acids, −ve = negatively charged amino acids, A = aromatic amino acids, S = amino acids with sulfur groups. Correlations higher than 0.80 are in bold numbers.

## Data Availability

Data is contained within the article.

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
