# Peer review of "Design of Plant-Based Food: Influences of Macronutrients and Amino Acid Composition on the Techno-Functional Properties of Legume Proteins"

_foods, 2023, doi:10.3390/foods12203787_

Round 1

Reviewer 1 Report

Page 2, 2.1 Materials: Why did the authors choose these legumes and these products for their goals? Please explain.

Page 2 , Table 1, Line 85: Maybe it is better to change ‘1P, LPI = pea, lupin; P, F, CPC = pea, fava bean, chickpea protein concentrate’ by ‘PPI = pea protein isolate, LPI = lupin protein isolate, PPC = pea protein concentrate,  FPC = fava bean protein concentrate, CPC = chickpea protein concentrate’ Change this in the all tables.

Page 4, 139: Please change ‘amnio acids’ by amino acids.

Page 5, Figure 1: Why did the authors present EAI and ES in a common diagram (see 1C)? Maybe it is better to divide it into two diagrams. 

Page 5, Line 161-184: Please indicate the results of the statistical tests on Figure 1 as well. Use asterisk (*) or letters (a, b) to represent a statistically significant result.

Page 5, line 162: Change ‘(5.89 g/g (± 0.22))’ by ‘(5.89 ± 0.22 g/g)’ avoiding using  brackets twice. Check this in the overall manuscript.

Page 4, Table 4: This table did not show correlations between functional properties and macronutrients. It shows correlations between functional properties and amino acid contents. Check the order of tables.

Why the authors wrote ‘H = hydrophobic amino acids, OH = amino acids with OH groups, P = polar amino acids, + = positively charged amino acids, - = negatively charged amino acids, A = amino acids with aromatic groups, S = amino acids with sulfide groups’ on the bottom of the table? Correlations between functional properties and these parameters are shown in Table 6. Please move them to the bottom of table 6.

Author Response

You can see the responses in the uploaded file.

Thank you

Reviewer 2 Report

This paper provides an in-depth analysis of the research on the relationship between amino acids, macronutrient composition and technical functional properties, which is of certain value. Below I listed some things that I consider important to be revised by the authors.

The logic of the Abstract part is not very clear. So many conjunctions are completely unnecessary. Please reorganize the Abstract part and the text according to the common logic.

Line 99: There must be Spaces between numbers and units. The text is not uniform.

Line 99: It is difficult to understand. How much protein, water and oil were added respectively? Please be more precise.

Line 114: What does Φ mean in the formula?

Line 156: The position "156" is not appropriate. Figure 1 is poorly presented and not clear.

Lines 163 - 166: There are very serious problems. Please be sure to verify the experimental results carefully to ensure the accuracy of the experiment.

Line 228: 4.2 should be changed to 4.1. The table in this paper is not standard. Please revised it.

Lines 244 - 246: I don't know why the WHC of PPI is the highest here? Does lupine protein isolate need no heat treatment?

Lines 248 - 249: Pea protein concentrate has a higher value than chickpea protein concentrate, what does this value refer to?

Line 254: The emulsion properties of proteins are particularly dependent on two factors: firstly, the rate at which the protein diffuses to the interface, and secondly, the deformability of its conformation under the influence of the interfacial tension (surface denaturation). Why was the interfacial adsorption kinetics of proteins not measured? It can more intuitively reflect the diffusion rate and adsorption capacity of proteins at the water-oil interface.

Lines 261 - 262: What does that mean?

Lines 267 - 269: Isn't the protein concentration of PPC lower than that of PPI? This statement confused me. Please be more specific.

Lines 274 - 275: The protein concentration of LPI is the highest. Why is its emulsification stability poor? This part should be deeply discussed.

Line 297: ":" should be deleted.

Lines 307 - 312: The part should be discussed in deep and references should be added to explain the result. The discussion should be enhanced to enhance the significance, with more recent literature.

Lines 373 - 375: Is this overturning the previous research results?

The English language need be revised, in several points it is hard to understand.

Author Response

You can see the Response in the uploaded file. Thank you

Round 2

Reviewer 2 Report

The manuscript has been revised to the level of publication.